# Humidity and Light Modulate Oxygen-Induced Viability Loss in Dehydrated *Haematococcus lacustris* Cells

**Thomas Roach** [1,*] **, Alessandro Fambri** [1] **and Daniel Ballesteros** [2,†]

1    Department of Botany, University of Innsbruck, Sternwartestraße 15, 6020 Innsbruck, Austria
2    Royal Botanic Gardens, Kew, Wakehurst Place, Ardingly RH17 6TN, UK
*    Correspondence: thomas.roach@uibk.ac.at
†    Current address: Department of Botany and Geology, University of Valencia, Av. Vicent Andres Estelles s/n, 46100 Burjassot, Spain.

**Abstract:** *Haematoccocus lacustris* (previously *H. pluvialis*) is a desiccation-tolerant unicellular freshwater green alga. During acclimation to desiccation, astaxanthin-rich lipid bodies and low-molecular-weight antioxidants ($\alpha$-tocopherol, glutathione) accumulate, while the chloroplast area and chlorophyll contents decrease, which may facilitate desiccation tolerance by preventing damage mediated by reactive oxygen species (ROS). Here, we investigated the influence of moisture, light, oxygen, and temperature on redox homeostasis and cell longevity. Respiration and unbound freezable water were detectable in cells equilibrated to $\geq$90% relative humidity (RH), a threshold above which viability considerably shortened. At 92.5% RH and 21 °C, antioxidants depleted over days as cells lost viability, especially in an oxygen-rich atmosphere, supporting the role of ROS production in uncoupled respiration in viability loss. At 80% RH and 21 °C, redox homeostasis was maintained over weeks, and longevity was less influenced by oxygen. Light and oxygen was a lethal combination at 92.5% RH, under which pigments bleached, while in the dark only astaxanthin bleached. Viability positively correlated with glutathione concentrations across all treatments, while correlation with $\alpha$-tocopherol was weaker, indicating limited viability loss from lipid peroxidation at 80% RH. In cells equilibrated to 50% RH, longevity and redox homeostasis showed strong temperature dependency, and viability was maintained at sub-zero temperatures for up to three years, revealing cryogenic storage to be an optimal strategy to store *H. lacustris* germplasm.

**Keywords:** *Haematococcus pluvialis*; desiccation tolerance; reactive oxygen species; antioxidants; oxidative stress; molecular mobility; redox metabolism; green algae; ageing



## 1. Introduction

Water is essential for cell activity. For example, water molecules bind to the primary structure of proteins and other polar macromolecules to enable their functional 3D structure [1]. Furthermore, when the cell's cytoplasm is sufficiently hydrated, molecular mobility and diffusion of large biomolecules occurs, permitting substrate transport for the vast majority of enzyme activities and metabolic processes [2,3]. While water is essential for life, some life forms have evolved to tolerate its absence. Aerial-terrestrial algae live in an environment of intermittent water supply and frequently face dehydration and desiccation. Dehydration causes organelles to shrink, and as cytoplasmic viscosity increases and molecular mobility decreases, enzymes have restricted access to substrates and cellular processes cease [2]. A suite of mechanisms is required to enable desiccation tolerance [4]. Once desiccated, the organism is in equilibration with atmospheric humidity, which subsequently dictates which metabolic processes can occur. Studies in seeds have shown that, at above 91% relative humidity (RH), respiration is possible, but does not operate normally, leading to elevated production of reactive oxygen species (ROS), which can cause macromolecular damage from processes such as lipid peroxidation [5,6]. While the cytoplasm is still fluid in

state, decreases in RH lead to an exponential increase in cytoplasmic viscosity. At 25 °C and below a threshold around 44–49% RH, the cytoplasm solidifies into a glassy state, enzyme activity completely ceases, and cell longevity extends [2,3,7]. Despite extended longevity in the glassy state, relative to the fluid state, cells are not devoid of ROS-mediated damage [8,9]. Photosynthesis is another process that requires regulation by process such as non-photochemical quenching (NPQ) to prevent excessive amounts of ROS production [10], particularly during desiccation [11–13]. In dehydrating desiccation-tolerant algae, it is reported that photosynthesis is usually down-regulated before respiration [14]. Nonetheless, light absorbed by chlorophyll in desiccated organisms is a source of ROS that can accelerate viability loss of seeds and spores [15–17].

Desiccation tolerant organisms use a variety of antioxidants to metabolise ROS, including enzymes and low-molecular-weight (LMW) antioxidants. Activity of antioxidant enzymes, such as ascorbate/glutathione peroxidases, glutathione reductase, catalase and superoxide dismutase generally increase during dehydration across the diversity of desiccation tolerant organisms, including yeast, tardigrades, nematodes, rotifers, seeds and leaves of 'resurrection' angiosperm plants [12,18–22]. However, once dehydrated, enzyme activity is inhibited and reliance of redox homeostasis lies with LWM antioxidants, including tocochromanols (e.g., $\alpha$-tocopherol) and carotenoids (e.g., astaxanthin) in the lipid domain (membranes and lipid bodies), and the tripeptide thiol glutathione (GSH) in the aqueous domain [8,23,24]. Loss of redox homeostasis closely associates with the loss of viability in desiccated seeds, lichens and leaves of resurrection plants [8,24,25]. However, the physical state of the cytoplasm in response to particular RH/temperature combinations have profound influences on which redox defenses are challenged during the storage of desiccation tolerant seeds, as part of the ageing process that eventually lead to viability loss. Generally, in the glassy state (i.e., <50% RH, <45 °C), GSH and tocochromanol depletion indicates that lipid peroxidation contributes to viability loss [9,26–28], whereas with fluid cytoplasm, but with inactive respiration (i.e., 50–91% RH, >5 °C), tocochromanol levels remain stable, whereas GSH depletion still occurs [9,27,29–31]. Desiccation-tolerant seeds equilibrated to 100% RH and >0 °C have a fluid cytoplasm, active respiration (thus age very rapidly) and lipid peroxidation again can associate with viability loss [32,33]. Lipids may play another role in longevity. While deterioration of biological material tends to slow as the storage temperature decreases [2,34,35], some seeds and fern spores exhibit anomalous longevity responses to temperatures between 10 and −30 °C [36]. This anomaly has been linked to triacylglycerol (TAG) crystallization/melting kinetics of storage lipids at specific storage temperatures [35,37–39].

*Haematococcus lacustris* (previously called *H. pluvialis*) is a desiccation-tolerant Chlorophyta (green algae), which is well known for its production of the ketocarotenoid and astaxanthin when subjected to stress [40]. Astaxanthin has gained commercial interest, partly because it is one of the most efficient antioxidants inhibiting lipid peroxidation [41,42]. During acclimation to desiccation, *H. lacustris* synthesises astaxanthin-rich lipid bodies that accommodate a large fraction of cell volume, accumulates $\alpha$-tocopherol and GSH several fold, reduces chloroplast area and chlorophyll levels; changes that all reverse within days of rehydration [23]. The acclimation process of liquid-cultivated cells required atmospheric exposure for a few days in a humid atmosphere, and light to drive photosynthesis, but not necessarily high light stress [23]. Without acclimation, many liquid-grown cells collapsed during dehydration, while those that remained intact did not recover photosynthetic activity during rehydration, showing a lack of desiccation tolerance. Lipid body synthesis can have a structural role in preventing cellular collapse during dehydration, while antioxidant accumulation and down-regulating photosynthesis may relate to preventing ROS production and a loss of redox homeostasis. Therefore, changes during desiccation acclimation may form part of a strategy to not only survive the physical demands of desiccation, but also give cells the maximum chance of survival (so-called 'longevity') in the dehydrated state until they are able to resume metabolism after rehydration.

Here, we have investigated some of the factors that cause viability loss of dehydrated *H. lacustris* cells. In the first experiment, we aimed to investigate the temperature dependency of ageing reactions in *H. lacustris* dry cells and elucidate whether anomalous longevity responses are found at the temperatures in which the abundant cell's storage lipids crystallize and melt. Therefore, cells were equilibrated to 50% RH and 20 °C, (conditions that lead the cytoplasmic aqueous phase to form a glassy state) before storage in the dark for up to three years at temperatures between −80 °C and 20 °C. In the second experiment, we investigated how the cytoplasmic physicochemical state affects the interaction of oxygen tension with light in mechanisms of ageing. To elucidate how cellular redox homeostasis supports cell longevity, concentrations of LMW antioxidants were correlated with the viability of cells aged at 80.0% and 92.5% RH, which corresponded to the two contrasting cytoplasmic physicochemical states based on the absence and presence of unbounded/freezable water, respectively. Overall, the results of this paper expand our knowledge of the fundamentals of desiccation tolerance and longevity in a greater diversity of plant systems. In addition, they provide insights into the optimal conditions for the long-term storage and conservation of *H. lacustris* genetic resources.

## 2. Materials and Methods

### 2.1. Algae Cultivation, Dehydration and Rehydration

*H. lacustris*, purchased as *H. pluvialis* strain 34-1b from the Culture Collection of Algae (SAG) at the University of Göttingen, was cultivated in 500 mL Schott bottles (Duran) filled approximately to 200 mL with 3N-bold basal medium (BBM) media. The cultures were bubbled from silicone tubing with air provided by an aquarium pump, fitted with 0.22 μm filters at the inlet and outlet, through GL14 screw-cap lids. Liquid cultures were placed under continuous light from a compact fluorescent tube (Envirolite; 200 W-6400 K) at an intensity of 50 μmol quanta $m^{-2}$ $s^{-1}$. The room was temperature controlled to 21 °C. After 1 week, cells were transferred onto 5 μm pore-size cellulose acetate filters (Sartorius) with a Büchner flask using gentle vacuum to remove media, and slowly dehydrated at 20 °C over 7 days on moistened Whatman filter paper, according to Reference [23]. Two filter sizes were used: 2 cm diameter for pigment and tocopherol measurements and 4.5 cm diameter for GSH. Before transferring cells, filters were freeze-dried and weighed to obtain the dry weight (DW) without biomass. For full rehydration, filters were saturated with 3N-BBM media and placed under 50 μmol quanta $m^{-2}$ $s^{-1}$ at 21 °C.

### 2.2. Differential Scanning Calorimetry

The physicochemical status of the cells under contrasting RH and temperatures was measured via the cells' thermal behavior using differential scanning calorimetry (DSC). These measures allowed us to select conditions for the long-term storage treatments initially, but also provided a physicochemical and structural reference for the interpretation of the biochemical and physiological measures during the storage experiments. First and second order transitions of water and storage lipids (i.e., TAG) were determined using a Mettler-Toledo DSC-1 (Greifensee, Switzerland), calibrated for temperature and energy with indium standards (156.6 °C and 28.54 J $g^{-1}$, respectively). 3–10 mg of cells were sealed into 40 μL Mettler-Toledo DSC pans that were cooled from 25 °C to −150 °C at a rate of 10 °C $min^{-1}$, held isothermally for 1 min, and then heated from −150 °C to 90 °C at a rate of 10 °C $min^{-1}$ [43]. Thermal events were assessed during cooling and warming scans. Enthalpy (ΔH) of first order transitions (i.e., crystallization, melting or recrystallization) was calculated from the area encompassed by the peak and the baseline and the onset temperature was determined from the intersection between the baseline and a line drawn from the steepest portion of the transition peak. Second order transitions in heating scans were identified as glass melting events, and the glass transition temperature (Tg) was assigned as the midpoint in the displacement of power during the scan, as previously performed for other unicellular plant systems [35,44]. All analyses were performed using

Mettler-Toledo Stare software version 12.0, Mettler-Toledo, Zurich, Switzerland. Enthalpies of exothermic and endothermic events are expressed on a $g^{-1}$ DW basis.

### 2.3. Long-Term Storage at Various Temperatures and 50% RH in the Dark

For storage experiment 1, desiccated cells on filters were equilibrated to 50% RH, 20 °C before sealing three filters (replicates) together in tri-laminated aluminium foil bags (light, moisture and gas exchange proof for the length of the experiment) while in the RH chamber, which were stored at diverse temperatures: 20 °C, 4 °C, −20 °C, −50 °C, and −80 °C. At each time point, one envelope was removed for measurement.

Longevity of the cells stored under different conditions was calculated from changes in the maximum quantum yield of photosystem II ($F_v/F_m$, detailed below) (Nt) with storage time (t). Longevity was expressed as P50 (time for initial $F_v/F_m$ to decline to half that value) and was calculated by the Avrami function [35,45]. Aging rate was expressed as the reciprocal of P50 (P50$^{-1}$). For the Avrami function ($\ln(N_t/N_0) = n \cdot (t/\phi)$), where $N_0$ = initial or maximum $F_v/F_m$ for the sample), a linear regression of $\ln(t)$ and $\ln[\ln(N_t/N_0)]$ was used to calculate Avrami coefficients: n (slope of linear regression), $\phi$ [$\phi = \exp(-y_0/n)$ and $y_0$ = the y-intercept of linear regression] [35]. Values for $N_0$ for *H. lacustris* cells were calculated from the average of the five highest $F_v/F_m$ measurements across all treatments. Values for $N_t/N_0$ were constrained between 0.995 and 0.001 (e.g., Reference [35]). P50 could be calculated by the interpolation when $F_v/F_m$ declined below 50% of the original during the course of the experiment (P50 < 1100 days (3 years)). However, we needed to extrapolate models when insufficient deterioration occurred [35].

The effects of temperature on longevity were characterized using classic Arrhenius plots where the rate = $A_0 \cdot \exp^{(-Ea/RT)}$, with T = temperature (in Kelvin), R = 8.314 J K$^{-1}$mol$^{-1}$ (the ideal gas constant), $A_0$ = a pre-exponential factor, and Ea considered the apparent activation energy or temperature coefficient [35,46,47]. Apparently linear relationships of $\ln(P50^{-1})$ and $T^{-1}$ were used to estimate Ea in order to make inferences about the temperature dependency of the aging rate and whether there were deviations from linearity at specific temperatures. Linear relationships were calculated using Excel and an F distribution model with a probability set at 0.05.

### 2.4. Short-Term Storage at 21 °C, Various RH, in Light or Dark and Contrasting Oxygen Tension

For storage experiment 2, desiccated cells on filters were placed in 1–1.5 L glass jars, each containing 3 mL of a non-saturated lithium chloride solution placed in a petri dish (4 cm diameter), between 30% and 99% RH, at 21 °C, as measured with Rotronic HC2-AW-USB probe. Jars were either flushed with pure oxygen ($O_2$) or nitrogen ($N_2$) gas for 20 s at a flow rate of 50 mL s$^{-1}$. All jar seals were greased before tightening the lids to ensure they were hermetic. The average $O_2$ saturation in the jar flushed with $O_2$ and $N_2$ was 89.8% and 1.1%, respectively, as measured with the Fibox 3 Minisensor Oxygen Meter (PreSens) via an Oxygen Sensor Spot (PreSens) inside every jar. Jars were either placed under the light intensity used for growing cells, or in a lightproof box, as stated in the results, at 21 °C. The short-term storage experiment was repeated twice with the same time intervals of 0, 7, 14, and 25 days. During the first series, only chlorophyll fluorescence ($F_v/F_m$, see below) was measured to calculate longevity expressed as P50 via the Avrami function (see above). For the second series, pigments and antioxidants were measured.

### 2.5. Sample Preparation for Biochemical Analyses

Cells were frozen in liquid nitrogen, briefly stored at −80 °C and freeze-dried under a vacuum of 0.04 bar for 72 h. After drying, filters and cells were weighed, and filter DW was subtracted to calculate cell DW. For GSH and pigment measurements, half of a big filter and an intact small filter was used, respectively. Filters were roughly cut into fine pieces with scissors and cells were ground on filters inside 2 mL Eppendorf reaction tubes and with two 5 mm agate beads with a pre-chilled Tissue Lyser Block (−80 °C) for 2 min at a frequency of 30 Hz.

### 2.6. HPLC Analyses of Pigments, Tocopherols and Glutathione

HPLC methods were conducted, as in Reference [23]. Pigments and $\alpha$-tocopherol were extracted in 600 μL methanol and measured simultaneously from the same 10 μL injection volume after separation on a LiChrospher 100 RP-18 column (125 × 4 mm, 5 μm), via absorbance (440 nm for carotenoids and 650 nm for chlorophyll) and fluorescence (Ex: 295 nm, Em: 325 nm), respectively. For measuring the total amounts of each LMW thiol/disulphide couple, extraction was in 1.0 mL 0.1 N HCl, which was the pH adjusted to 7.0 with 200 mM bicine buffer, pH 8.8, for reduction of disulphides to thiols with dithiothreitol, and thiol labelling with mono-bromobimane. Labelled thiols in a 10 μL injection volume were detected via fluorescence (Ex: 380 nm, Em: 480 nm) after separation on a ODS Hypersil column (250 × 4.6 mm, 3 μm), and quantified with pure standards (Sigma, St. Louis, MO, USA).

### 2.7. Statistical Analyses of Biochemical Measurements

The data shown are the average of three biological replicates, each composed of cells on separate filters. For Experiment 1 (long-term storage), significant differences were calculated via ANOVA across storage temperatures. For Experiment 2 (short-term storage), significant differences relative to desiccated cells before ageing were calculated with a *t*-test, assuming asymmetric distribution of data.

### 2.8. Chlorophyll Fluorescence Analyses

Measurements of chlorophyll fluorescence of rehydrated cells were conducted with an IMAG-K6 CCD camera (M-series, Walz, Pfullingen, Germany) of a few cells from the same filter used for biochemical measurements. Additional images were made with a modified Axio Scope.A1 (Zeiss, Göttingen, Germany) epifluorescence microscope attached to the IMAG-K6 CCD camera. Cells were rehydrated on filters in 3N-BBM under continuous growth light for 48 h before measurements. For the determination of maximum quantum yield of photosystem II (PSII) ($F_v/F_m$), as calculated via $(F_m{}^\circ - F_o)/F_m{}^\circ$, a 600 ms saturating pulse was applied to measure $F_m{}^\circ$ (maximum fluorescence). All samples were dark-adapted for 0.5 h before measurement. Desiccated cells were measured with a FluorCam 701MF CCD camera (Photon System Instruments, Drásov, Czech Republic).

## 3. Results

### 3.1. Physicochemical Status of H. lacustris Cells Based on DSC

Data from DSC of desiccation tolerant cells equilibrated to water contents (WC) between 0.017 and 0.594 g $H_2O$ $g^{-1}$ DW showed that increases in the WC led to increases in enthalpy of the melting peaks ($\Delta H_m$) (Figure 1a), which is indicative of a greater amount of water that is able to freeze and melt. The intersection of the linear regression between WC and $\Delta H_m$ revealed a threshold WC of 0.17 g $H_2O$ $g^{-1}$ DW, corresponding to an equilibration RH > 90%, before unbound water was freezing/melting (Figure 1b). This WC threshold was also apparent in non-desiccation tolerant *H. lacustris* cells equilibrated to RH between 90–94%, and corresponding with an average WC of 0.21 g $H_2O$ $g^{-1}$ DW (data not shown). Therefore, in this study, we used cells equilibrated to three RH atmospheres, each leading to contrasting physicochemical statuses to investigate the impact of the environment on the physiology of dehydrated cells: 50% RH for long-term storage of cells with a glassy cytoplasm (Experiment 1), and 80.0% and 92.5% RH (without and with unbound water, respectively) for the short-term storage of cells with a fluid cytoplasm (Experiment 2).

Fully desiccated desiccation-acclimated cells of *H. lacustris* (e.g., <50% RH) showed a complex profile of first order transitions (peaks) from 20 to $-60$ °C (Figure S1). These occurred at the same temperatures and with similar magnitudes (i.e., $\Delta H_m$) in cells with diverse WC, indicating that they correspond to the crystallization and melting of TAG storage lipids. This was further supported by an absence of peaks in non-desiccation-acclimated cells (not shown) that contain hardly any lipid bodies [23]. Melting peaks spanned between temperatures of 17 °C to 0 °C, 0 to $-22$ °C, and $-25$ °C to $-60$ °C

(Figure S1). Based on this thermal profile, temperatures above and below these peaks (20 °C and −80 °C, respectively), and within the range of the lipid melting peaks (4 °C, −20 °C and −50 °C), were used for the long term storage of cells in Experiment 1. The focus here was placed on the temperature dependency of redox-associated ageing reactions in long-term stored dry cells, and whether lipid crystallisation and melting events at the storage temperatures could have potentially negative effects on longevity.

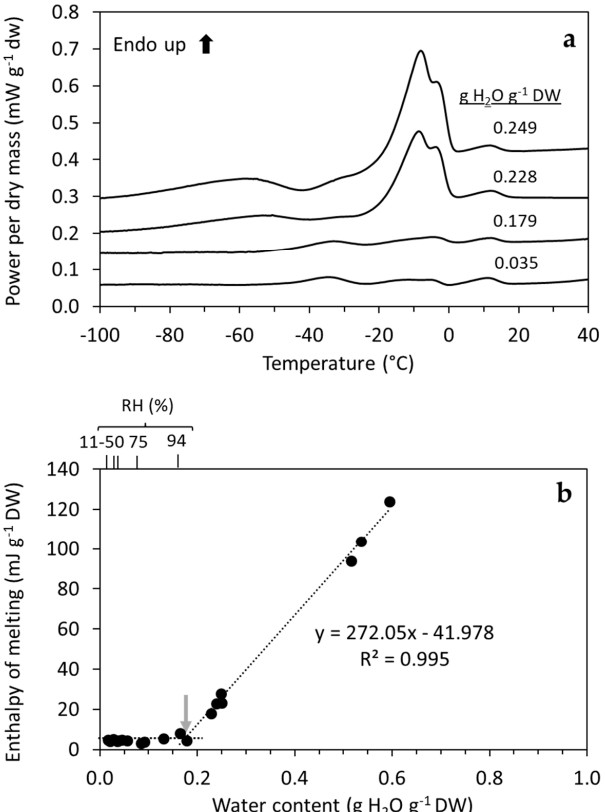

**Figure 1.** Influence of temperature and water content on ice melting events detected via DSC. (**a**) Effect of WC on the profile of first order transitions (peaks) in heating DSC scans from −100 °C to 40 °C of *H. lacustris* cells. For clarity, scans from cells >0.035 g H2O g$^{-1}$ DW have been stacked above one another, thus Y-axis values are not absolute. (**b**) Relationship between water content and enthalpy of melting (i.e., peak size in (**a**)), revealing that unbound and freezable water only occurred in cells above 0.171 g $H_2O$ g$^{-1}$ DW (grey arrow).

### 3.2. Temperature Dependency of Ageing Reactions in Desiccated H. lacustris Cells

Experiment 1 showed that temperature had a clear effect on cell longevity. After 122 days at 50% RH in the dark and ambient oxygen, $F_v/F_m$ values of rehydrated cells previously stored at 20 °C decreased, while after 242 days they were 0, indicating a complete loss of viability (Figure 2a). After 242 days, storage at 4 °C, $F_v/F_m$ of rehydrated cells had started to decline, which further decreased after 365 days, at the time point at which $F_v/F_m$ values of cells stored at sub-zero temperatures had not yet changed (Figure 2a). Cells equilibrated to 50% RH contained no freezable water (Figure 1b). Microscopic imaging of chlorophyll fluorescence at the cell level revealed consistent $F_v/F_m$ values across the population 48 h after rehydration (Figure 2b).

Fitting the experimental data into the Avrami function, the time for 50% loss of initial $F_v/F_m$ values (P50) was calculated to be 148 and 383 days for cells stored at 20 °C and 4 °C, respectively. Loss of $F_v/F_m$ under these conditions was significant when data was analyzed by Generalized Linear Models using logistic (sigmoidal) functions (GLM, $p < 0.01$). P50s for cells stored at −20 °C, ≤−50 °C were estimated to be 822 and >1 million years, respectively, based on extrapolations of the Avrami function (Figure 2a; Table 1). These estimations provided

very large numbers as $F_v/F_m$ values in the last viability assay (3 years) were similar to those taken at time zero or after 1.1 years of storage, with <10% change. Ageing at these conditions was not significant over the three years when data was analyzed by GLM ($p > 0.05$).

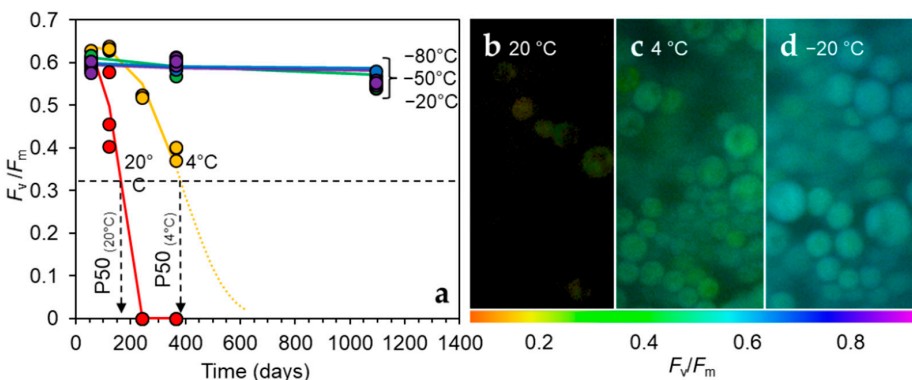

**Figure 2.** Effect of temperature on longevity of cells equilibrated to 50% RH before storage in aluminum foil wraps. (**a**) Chlorophyll fluorescence-derived $F_v/F_m$ values after 48 h of rehydration, used as a cell viability indicator. $F_v/F_m$ values from (**a**), but as images of individual cells using a false-color scale (below) after 1 year in storage at (**b**) 20 °C, (**c**) 4 °C, and (**d**) −20 °C.

**Table 1.** Effect of storage conditions on *H. lacustris* cell longevity (P50).

| Temperature (°C) | RH (%) | O$_2$ (%) | Light | P50 (Days) | P50 (Years) |
|---|---|---|---|---|---|
| 20 | 50 | 21 | Dark | 148 | 0.4 |
| 4 | 50 | 21 | Dark | 383 | 1.1 |
| −20 | 50 | 21 | Dark | 299,885 * | 821.6 * |
| −50 | 50 | 21 | Dark | $3.8 \times 10^{12}$ * | $10 \times 10^9$ * |
| −80 | 50 | 21 | Dark | $1.7 \times 10^{12}$ * | $4.7 \times 10^9$ * |
| 21 | 80 | 90 | Dark | 41.8 | <0.1 |
| 21 | 80 | 1 | Dark | 40.2 | <0.1 |
| 21 | 80 | 90 | Light | 21.4 | <0.1 |
| 21 | 80 | 1 | Light | 37.9 | <0.1 |
| 21 | 92.5 | 90 | Dark | 7.3 | <0.1 |
| 21 | 92.5 | 1 | Dark | 17.2 | <0.1 |

* P50s extrapolated from Avrami equations.

The profound effect of temperature on the ageing rate (P50$^{-1}$) in dry *H. lacustris* cells was also characterized using Arrhenius plots. A significant linear relationship was observed for the full range of storage temperatures of the experiments that ranged between 20 °C and −80 °C ($p = 0.023$, $r^2 = 0.86$), although the correlation coefficient was higher when the range between 20 °C and –50 °C was considered ($p = 0.021$, $r^2 = 0.96$) (Figure S2). Slopes for these temperature ranges were −15 °C (including −80 °C) and −23 °C (excluding −80 °C), corresponding to an apparent activation energy (Ea) of 126 and 192 kJ mol$^{-1}$, respectively (Figure S2). Faster than expected aging rates for spores stored at specific temperatures would have been detected as a break in the linear behavior of the Arrhenius plot [35], but this was not the case. Differences in slope coefficients due to the similar ageing rate at −50 °C and −80 °C may be an artefact of the calculations because no real changes in $F_v/F_m$ were observed during this relatively short storage time (three years). Hence, the value of the calculated Ea must be confirmed by future assessments.

Cells at 50% RH that were stored for one year at 20 °C had depleted the total GSH (Figure 3a) and much lower α-tocopherol levels (Figure 3b), as compared to those stored at −20 °C or colder, whereas cells stored at 4 °C had lower, but non-significant levels of

either antioxidant compared to sub-zero storage. Regarding carotenoid amounts relative to total chlorophyll, only total violaxanthin, antherxanthin and zeaxanthin (VAZ) levels were significantly lower in cells stored at 20 °C, relative to colder temperatures (Figure 3c). While on a DW basis astaxanthin levels were lower in cells stored at 20 °C, there was no change in astaxanthin levels relative to total chlorophyll (Figure 3d), i.e., rates of chlorophyll and astaxanthin loss were equal. After three years, there was a slight, but insignificant decrease in $F_v/F_m$ values of rehydrated cells previously stored at sub-zero temperatures.

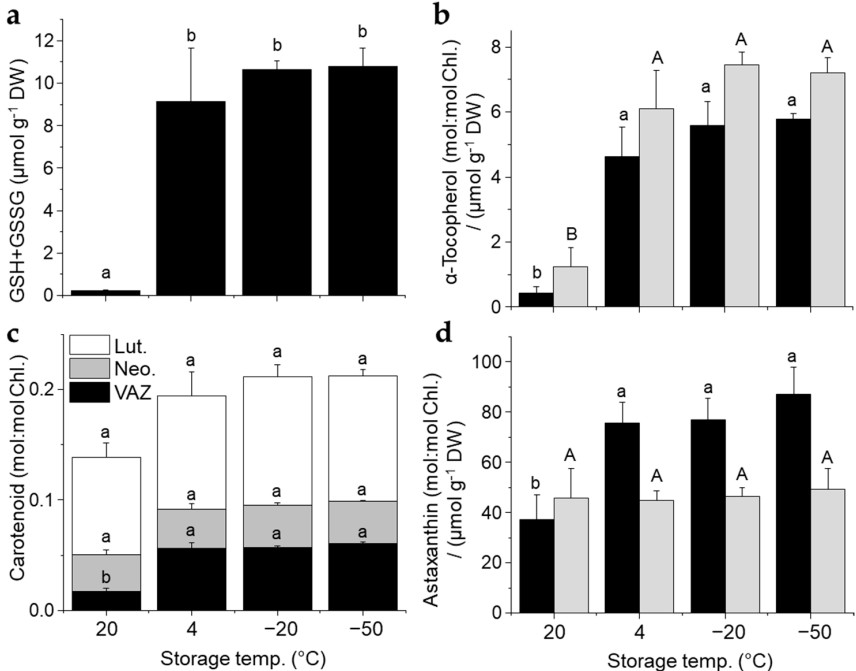

**Figure 3.** Effect of temperature on LMW antioxidant and pigment concentrations in cells equilibrated to 50% RH before storage in aluminum foil wraps for one year. Concentrations of (**a**) total glutathione, (**b**) α-tocopherol, (**c**) carotenoids, and (**d**) astaxanthin. Data in (**b**,**d**) is normalized on dry weight (DW) basis (black bars), and on the same scale, normalized to total chlorophyll (Chl., grey bars). Different lower case and capital letters denote significant differences ($p < 0.05$) when normalized to DW and Chl., respectively, $n = 3 \pm$ SD. Lut. = lutein; Neo. Neoxanthin, VAZ = xanthophyll cycle pool.

### 3.3. Oxygen and Light-Dependent Effects on Viability Loss, Pigment Bleaching, and Antioxidant Metabolism at High RH and Contrasting Physicochemical Status

Experiment 2 showed that oxygen and light had different effects on viability loss, oxygen consumption and pigment bleaching depending on the physicochemical status of the cells provided by high storage RH. After 20 days under ambient oxygen and light, full recovery of $F_v/F_m$ occurred in cells stored >80% RH, while at RHs > 92.5% it did not and cells bleached, showing complete inviability (Figure 4). No oxygen consumption was detected between 30% and 80% RH (Figure 4a), while with an increasing amount of oxygen consumed, there was a higher storage of RH, thus suggesting an increasingly active respiration (Figure 4a).

During storage at 80.0% RH, viability loss was less influenced by oxygen, but accelerated by light (Figure 4b) with P50s that ranged between 40.2 and 41.8 days for dark stored cells and between 37.9 and 21.4 days for light stored cells in high and low oxygen tensions, respectively (Table 1). At 92.5% RH, viability was rapidly lost at high oxygen tension, irrespective of light with P50s between 7.0 and 7.3 days, while light accelerated viability loss in a low oxygen tension (Figure 4b, Table 1). After 4–8 weeks at ≥92.5% RH under a high oxygen tension in the dark, bleaching of astaxanthin, but not chlorophyll, occurred (Figure 4c), a process that had already started to occur within 7–14 days (Figure S3). At <92.5% RH, bleaching of astaxanthin under a high oxygen tension was not observed (Figures 4c and S2).

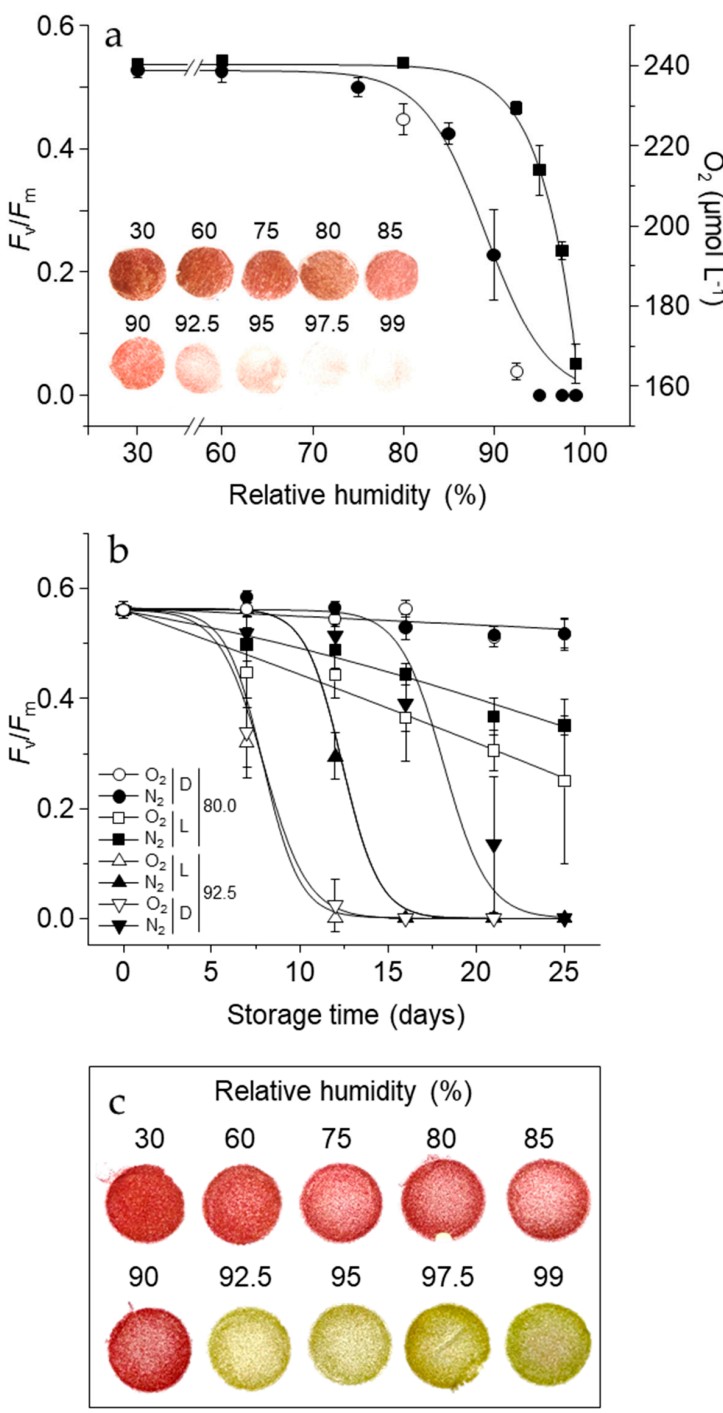

**Figure 4.** Interaction of oxygen, light and RH on cell longevity and pigment bleaching. $F_v/F_m$ values after 48 h rehydration, as a viability indicator, of cells stored for (**a**) 20 days in light at various RH, and (**b**) up to 25 days at 80.0% or 92.5% RH in dark (D) or LL and in high oxygen tension ($O_2$; open symbols) or low oxygen tension ($N_2$; closed symbols), $n = 4 \pm$ SD. Open circles in (**a**) denote the RHs used in (**b**). Squares in (**a**) show oxygen concentrations (right Y axis) of the atmosphere surrounding dark hermetically stored cells after 20 days. The inset in (**a**) shows typical filters after four days rehydration, with pigment bleaching indicative of viability loss. (**c**) Images of dry cells on filters after eight weeks storage at various RH in the dark and under a high oxygen tension.

Regarding the effects of light and oxygen tension on antioxidant and pigment changes at 80.0% and 92.5% RH, the overall trend was that depletion occurred more at 92.5% RH and this was accelerated by light and oxygen (Figure 5). Changes in GSH concentrations

associate closely with losses in viability. For example, at 92.5% RH, GSH depleted rapidly, whereas at 80.0% RH, loss of GSH occurred quicker in the light, but was not consistently affected by oxygen tension (Figures 5 and S4). GSH concentrations and $F_v/F_m$ values of rehydrated cells positively correlated with $R^2$ values of 0.80 and 0.45, under high and low oxygen tensions, respectively (Figure S4). Concentrations of γ-glutamyl-cysteine, the dipeptide precursor of GSH, increased from 0.2 to 1.5 μmol g$^{-1}$ DW over seven days at 92.5% RH under a low oxygen tension in the dark (Figure S4), despite GSH concentrations not increasing (Figure 5). However, with longer time, and similar to GSH concentrations, γ-glutamyl-cysteine concentrations rapidly fell as an increasing number of cells lost viability (Figure S4). In contrast, at 80.0% RH, γ-glutamyl-cysteine concentrations remained stable. Likewise, at 80.0% RH, levels of lipid soluble antioxidants remained relatively stable (Figure 5). Depletion of α-tocopherol and astaxanthin only occurred under light and a high oxygen tension at 92.5% RH (Figure 5), despite viability loss also occurring under low oxygen tension in the dark (Figure 5b). Therefore, $F_v/F_m$ values of rehydrated cells and cellular α-tocopherol and astaxanthin concentrations under a low oxygen tension only weakly correlated, with $R^2$ values levels of 0.29 and 0.12 for each compound, respectively, while under a high oxygen tension the correlation was stronger with $R^2$ values of 0.49 and 0.50, respectively (Figure S5).

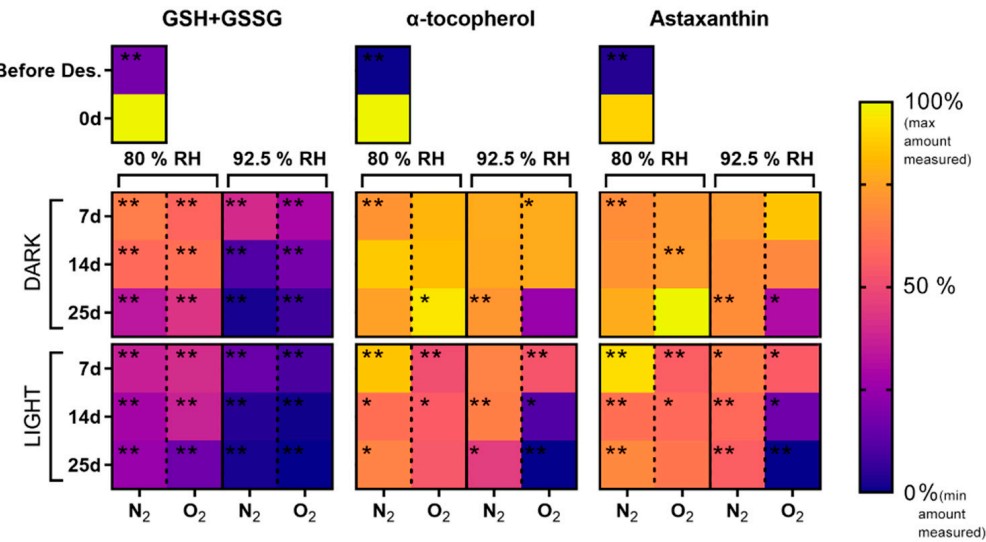

**Figure 5.** Interaction of oxygen, light and RH on changes in antioxidant and pigment concentrations in dehydrated cells. Data show relative differences on a false color scale (right side) of cells before desiccation (before Des.), immediately after desiccation (0 d), or after 7, 14 and 25 days (d) at 80.0% RH and 92.5% RH, under dark (upper panel) or light (lower panel), and in a high ($O_2$) or low ($N_2$) oxygen tension (below). Data have been normalized to dry weight and have shown relative to maximum values measured. Significant differences at $p < 0.05$ and $p < 0.01$ of absolute amounts between 0 d and all other time points are represented with * and **, respectively.

### 3.4. PSII Activity in Dehydrated Cells Equilibrated to Contrasting RH

Next, we used chlorophyll fluorescence to probe if PSII activity was active at 92.5% or 80.0% RH. The marginally higher $F_v/F_m$ and recoverable light-induced fluorescence quenching at 92.5% indicated that PSII in cells was able to perform minor levels of charge separation, leading to limited electron and proton transport (Figure 6a). After two weeks equilibration to various RH and four hours of rehydration, chlorophyll fluorescence traces again revealed differences in cell response to RH. For example, after storage at 92.5% RH a lack of light-induced quenching indicated an inability of cells to perform NPQ (Figure 6b). Furthermore, the rise in fluorescence in cells stored at 17–80% RH that occurred immediately after the actinic light was switched off was not apparent in cells stored at 92.5% RH (Figure 6b).

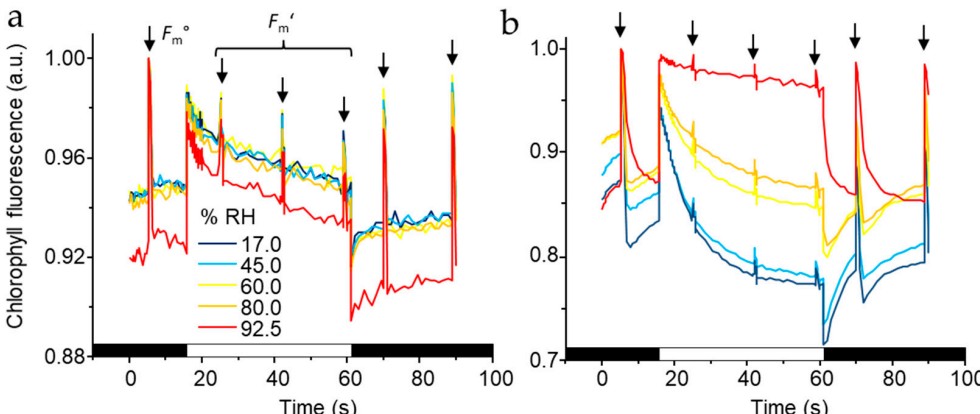

**Figure 6.** Chlorophyll fluorescence kinetics revealing effects of contrasting RH on photosynthesis. Traces are of (**a**) dry cells equilibrated to various RH, as indicated in the legend, for two weeks under light and ambient oxygen, and (**b**) after four hours re-hydration. Arrows in (**a**) indicate saturating pulses of light, given at the same time in (**b**), during the dark ($F_m°$, first measurement) or during illumination at 200 μmol photon m$^{-2}$ s$^{-1}$ ($F_m'$). Black and white bars on the X-axis indicate dark and illumination, respectively. Data are normalized to $F_m°$ and are averages of three filters.

## 4. Discussion

Desiccation tolerance is trait that has evolved across the diversity of life as an adaption to sporadic water availability, and to aid the dispersal of spores, pollen and seeds of non-desiccation tolerant organisms. The algal specie *H. lacustris* inhabits temporary water bodies that periodically dry, and desiccation tolerance enables dry cells to rapidly flourish after water returns. We have previously shown that, during acclimation to desiccation, the alga synthesises astaxanthin-rich lipid bodies, accumulates antioxidants, and down-regulates photosynthesis [23], all features that may relate to tolerating desiccation [4,8,12,14]. Here, we used specific treatments, involving light and dark, contrasting oxygen tension, RH, and temperature, designed to challenge desiccated cells in various ways, in order to investigate under which conditions the components that changed during desiccation acclimation supported longevity.

During acclimation to desiccation, the lipid composition alters significantly due to breakdown of thylakoid membranes and accumulation of TAG-rich lipid bodies [23]. Consequently, the percentage of polyunsaturated fatty acid (PUFA) linonelic acid (C18:3) of total FA decreases, while actual levels of almost all FAs, including linoleic (C18:2), oleic (C18:1) and palmitic (C16:0) acid increases. In the DSC melting scans (Figure S1), each distinctive peak can potentially be used as indicative of the melting of a different TAG. Based on the literature [48], the peak at about −35 °C is estimated to correspond to TAG mainly formed of C18:3, while the broad peak between −10 °C and 0 °C may correspond to C18:2 and C18:1. The peak around 11 °C may correspond to a combination of saturated FA of 10-16 carbons, as can be seen in *Cuphea lutea* melting thermograms [49]. Overall, major FA in desiccated *H. lacustris* cells are C18:2 (31%), C18:1 (23%), C16:0 (21%), C18:3 (14%), C14:4 (5%), C16:3 (2%), and C18:0 (2%) [23]. Since this represents whole cells, part of the C18:3, C18:2 and C16:0 fraction is from galactolipids of thylakoid membranes. The lipid composition of major TAGs in extra-plastid lipid bodies of *H. lacustris* has been shown to be mostly composed of variations of C18:1, C18:2 and C16:0, but also includes a combination of any of the aforementioned FA [50].

Crystallization and melting occur to TAG at specific temperature windows, leading to volume changes in the lipid droplets that, in dry cells (where the lipid droplets are surrounded by a glassy aqueous matrix), may promote instability of the glassy matrix [51]. This has been hypothesized as the mechanism to explain the shortened longevity of some seeds and fern spores during storage at some sub-zero temperatures [35,37–39,51]. The lipid peaks detected with DSC (Figure S1) represent such crystallization and melting

events. Using a variety of temperatures for long-term storage of desiccated cells (Figure 2), we explored, in Experiment 1, the potential for lipid crystallization and melting kinetics hindering cell longevity. However, this was not the case in *H. lacustris* cells. For example, P50s increased as temperature decreased (Table 1), and no anomalies were found for the temperatures studied in the three years of storage duration. Anomalies could have been expressed by a smaller P50 during storage at −20 °C compared to 4 °C or by smaller P50 during storage at 4 °C compared to 20 °C. These anomalies would have also been expressed as breaks in the linear trend of the Arrhenius plot [35], but there were no obvious breaks in this linear relationship. Indeed, what is revealed by the Arrhenius plot (Figure S2), is a very strong temperature dependency for longevity in *H. lacustris* cells, with an Ea = 192 kJ mol$^{-1}$ for the temperature range between 20 °C and –50 °C. This value is higher than other Ea values of diverse dry microbial and plant germplasm reported in this temperature range [2,34,35], but similar to viscosity measured in seeds based on structural relaxation considerations (185 kJ mol$^{-1}$ near Tg) [47]. The practical implications of these results is that dry sub-zero storage seems an excellent strategy to conserve *H. lacustris* germplasm for centuries, both at −20 °C (conditions of conventional seed banks) or cryogenic temperatures (either at −80 °C or in liquid nitrogen [as no viability loss was also found after one year of storage, data not shown]).

Due to its ability to form ROS, oxygen can be a major component in the deterioration of biological material. Storage of seeds with glassy cytoplasm under low oxygen tensions can significantly slow ageing rates [9,28]. Another component of cytoplasmic physical state is WC, as dictated by the RH material, which is equilibrated as well. At 25 °C, the glassy cytoplasm melts around 44–49% RH and increases in viscosity/fluidity with further increases in RH [5,7]. The role of oxygen in ageing processes becomes less clear in seeds with fluid cytoplasm [9,29–31,52], at least while respiration is inactive. Above 91% RH respiration is possible [5], and oxygen has an apparent role in ageing [32,33]. The detection of unbound/freezable water at ≥92.5% RH (Figure 1a), alongside the shortened longevity of cells under a high oxygen tension in Experiment 2 (Figure 4b; Table 1), supports that inefficient respiration that leads to elevated ROS production was the main source of rapid viability loss. Under such conditions, light had limited impact on longevity, indicating a limited contribution from ROS that can arise from chlorophyll and/or inefficient photosynthetic electron transport [12]. In support of this, chlorophyll did not bleach in cells stored in the dark at 92.5% RH that lost viability, despite extensive astaxanthin bleaching (Figure 4c and S3). When respiration was slowed by low oxygen tension, light had an obvious role in shortening longevity (Figure 4b; Table 1), indicating that ROS derived from inefficient respiration is inhibited at a lower oxygen tension than ROS derived from inefficient photosynthesis.

Chlorophyll fluorescence measurements showed slightly higher PSII activity in cells stored at 92.5% RH compared to 80.0% RH (Figure 6a), indicating that dehydration-induced inefficiencies in photosynthetic electron transport may have resulted in ROS production [10,53]. Moreover, shortly after rehydration, cells stored at 92.5% showed a lack of NPQ and transient post-illumination increase in chlorophyll fluorescence (Figure 6b), which is attributed to NAD(P)H dehydrogenase reduction of the plastoquinone pool [54,55]. Low levels of NAD(P)H, likely leading to low NAD(P)H dehydrogenase activity, would agree with the high oxidative load that cells had endured during time spent at 92.5% RH.

In summary, oxygen promoted deterioration at 92.5% RH, likely through ROS production from respiration, but also under a low oxygen tension from photosynthetic electron transport, inducing an intolerable oxidative load. The down-regulation of photosynthesis during acclimation to desiccation would help prevent ROS formation and associated deterioration in cells that naturally encounter high light intensities and are unable to perform efficient photosynthesis [23].

Ageing in the dehydrated state affected the cell's antioxidant defenses. The long-term storage (Experiment 1) of cells with glassy cytoplasm were under ambient oxygen, thus permitting ROS production. At 50% RH, the loss of viability at 20 °C was accompanied by

a loss of GSH, α-tocopherol and carotenoids (mainly xanthophylls, including astaxanthin) (Figure 3). However, antioxidant defences were differentially challenged at higher RH in cells with fluid cytoplasm (Experiment 2). GSH depleted faster at 92.5% than 80.0% RH and this was generally independent of oxygen tension, as shown before in seeds aged at high RH [9,29,30]. Interestingly, at 92.5% RH, the first enzyme-catalyzed step of GSH synthesis occurred, as shown by the increase in γ-glutamyl-cysteine concentration (Figure S4). GSH concentrations and redox state are reliable viability markers in various desiccation tolerant organisms [8], as well as light-stressed Chlorophyte alga [56]. The positive correlation of $F_v/F_m$ values of rehydrated cells, as used for cell viability, with GSH concentrations under all conditions (Figure S4) supports that this antioxidant associates with the maintenance of cell viability in the desiccated state of *H. lacustris*. This contrasts with the changes in lipophilic antioxidants, whereby α-tocopherol and astaxanthin concentrations only weakly correlated with viability in cells stored under a low oxygen tension that still lost viability (Figure S5). Therefore, other causes than lipid peroxidation likely contributed to viability loss, as is also the case for desiccation-tolerant seeds aged with fluid cytoplasm, but inactive respiration [9,27,29–31].

## 5. Conclusions

We conclude that the accumulation of LMW antioxidants during acclimation to desiccation is a mechanism *H. lacustris* uses to maximize survival chances, and this prevents the loss of redox homeostasis, while dehydrating, before rehydration and the reactivation of normal metabolism occurs. Antioxidant defenses deplete during ageing reactions, but the kinetics of the light and oxygen-induced redox imbalance and viability loss strongly depends on the physicochemical state of the cells. Thus, this is based on water availability and RH. While loss of GSH coincides with viability loss under any condition, lipid oxidation is strongly dependent on RH and oxygen availability. The lipid bodies in which α-tocopherol and astaxanthin may accumulate within are composed of diverse TAG, which each melt at temperature ranges between −60 °C and 17 °C. However, storage at any temperature within this range does not break the temperature dependency of longevity, thus supporting dry sub-zero storage for the long-term conservation of *H. lacustris* germplasm, especially under a low oxygen tension.

**Supplementary Materials:** The following supporting information can be downloaded at: https://www.mdpi.com/article/10.3390/oxygen2040033/s1, Figure S1. Profile of first order transitions in cooling and heating DSC scans of *H. lacustris* cells; Figure S2. Arrhenius plot describing the effect of temperature on the aging rate of *H. pluvialis* red cells stored at 50% RH; Figure S3. Effect of ageing in light dark, 80.0% and 92.5% RH and high and low oxygen tension on chlorophyll concentrations; Figure S4. Effect of ageing in light dark, 80.0% and 92.5% RH and high and low oxygen tension on glutathione and γ-glutamyl-cysteine concentrations, and correlation with viability; Figure S5. Effect of ageing in light dark, 80.0% and 92.5% RH and high and low oxygen tension on correlation of α-tocopherol and astaxanthin concentrations with viability.

**Author Contributions:** Conceptualization, T.R.; methodology, T.R. and D.B.; investigation, T.R., A.F. and D.B.; resources, T.R.; writing—original draft preparation, T.R., A.F. and D.B.; writing—review and editing, T.R. and D.B.; funding acquisition, T.R. All authors have read and agreed to the published version of the manuscript.

**Funding:** This research was funded by The Austrian Research Promotion Agency (FFG) BRIDGE programme (project ALAS, grant number 891093). Royal Botanic Gardens, Kew receives grant-in-aid from Defra, UK.

**Institutional Review Board Statement:** Not applicable.

**Informed Consent Statement:** Not applicable.

**Data Availability Statement:** Any raw data is available upon request.

**Acknowledgments:** We would like to thank Bettina Lehr (University of Innsbruck) for technical assistance with HPLC measurements.

**Conflicts of Interest:** The authors declare no conflict of interest.

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
