# Peer review of "Humidity and Light Modulate Oxygen-Induced Viability Loss in Dehydrated Haematococcus lacustris Cells"

_oxygen, doi:10.3390/oxygen2040033_

Round 1

Reviewer 1 Report

This is a very good paper, building on prior work by authors and collaborators on desiccation tolerance in Haematoccocus lacustris, herein assessing methods for best germplasm conservation/storage of this species by evaluating reasons for loss of viability under a range of storage environments.  I have only a few minor comments which I suggest need addressing. 

1. While it is clear that H. lacustris needs to be acclimated to achieve desiccation tolerance - this is based on previous work by the authors (ref 23).  In the current paper, the authors refer to comparison of data with non-acclimated cells, stating data not shown.  I feel it should be made clear to the reader at some point what the acclimation process requires and why it is necessary.  

2. I found an apparent contradiction in lines 320-321 "After 3 years, there was a slight decrease in Fv/Fm values of rehydrated cells previously stored at sub-zero temperatures", to what is stated in line 294 which states  “Ageing at these conditions was not significant over the 3 years..."

3. Minor typograpical errors can be found highlighted in yellow - as can comments about 1 and 2 above, in your article attached by myself. 

After 3 years, there was a slight decrease in Fv/Fm values of rehydrated cells previously stored at sub-zero temperatures Line 320-321.  This is in contradiction to the statement in lines 294 above which state  “Ageing at these conditions was not significant over the 3 years. 

Author Response

This is a very good paper, building on prior work by authors and collaborators on desiccation tolerance in Haematoccocus lacustris, herein assessing methods for best germplasm conservation/storage of this species by evaluating reasons for loss of viability under a range of storage environments.  I have only a few minor comments which I suggest need addressing. 

RESPONSE: Thank-you for reviewing our submission. We are happy you were positive about its content!

  1. While it is clear that H. lacustris needs to be acclimated to achieve desiccation tolerance - this is based on previous work by the authors (ref 23).  In the current paper, the authors refer to comparison of data with non-acclimated cells, stating data not shown.  I feel it should be made clear to the reader at some point what the acclimation process requires and why it is necessary.  

RESPONSE: Text was added in the introduction on lines 87-91: "The acclimation process of liquid-cultivated cells required atmospheric exposure for a few days in a humid atmosphere, and light to drive photosynthesis, but not necessarily high light stress [23]. Without acclimation, many liquid-grown cells collapsed during dehydration, while those that remained intact did not recover photosynthetic activity during rehydration, showing lack of desiccation tolerance." 

  1. I found an apparent contradiction in lines 320-321 "After 3 years, there was a slight decrease in Fv/Fm values of rehydrated cells previously stored at sub-zero temperatures", to what is stated in line 294 which states  “Ageing at these conditions was not significant over the 3 years..."

RESPONSE: We changed the sentence to "there was a slight, but insignificant, decreases in Fv/Fm" to  lines 320-321 

  1. Minor typograpical errors can be found highlighted in yellow - as can comments about 1 and 2 above, in your article attached by myself. 

 RESPONSE: We modified the text in the introduction referring to the first point highlighted in the pdf (see response above). The other typos were corrected.

After 3 years, there was a slight decrease in Fv/Fm values of rehydrated cells previously stored at sub-zero temperatures Line 320-321.  This is in contradiction to the statement in lines 294 above which state  “Ageing at these conditions was not significant over the 3 years. 

 RESPONSE: See our response above to this point.

Reviewer 2 Report

The manuscript entitled "Humidity and light modulate oxygen-induced viability loss in dehydrated Haematococcus lacustris cells" addresses a very relevant and appropriate topic for this journal. 

The manuscript is well structured, well-written and well-founded. Some minor fixes are suggested below.

Corrections needed:

line 26 - Keywords: Haematococcus lacustris, desiccation tolerance, reactive oxygen species, antioxidants, ...

line 46 - 25 °C, below a threshold around 44-49% RH, the cytoplasm solidifies into a glassy state,

line 98 - ... and melt. Therefore, cells were equilibrated to 50% RH at 20 °C, which corresponded

line 257 - above and below these peaks (20 and −80 °C) and within the range of the lipid melting

line 270 - 3.2. Temperature dependency of ageing reactions in desiccated H. lacustris cells (note: "H. lacustris " in italics)

line 284 - storage at (b) 20 °C, (c) 4 °C, and (d) −20 °C, on a false colour scale shown below the images.

line 288 - for cells stored at 20 and 4 °C, respectively. Ageing at these conditions was significant

line 308 - both –50 and –80 °C, leading to a slope with smaller correlation coefficient, may be an ...

line 393 - rapidly fell as viability was lost (Fig. S4). In contrast, at 80% RH, γ-glutamyl-cysteine

Author Response

The manuscript entitled "Humidity and light modulate oxygen-induced viability loss in dehydrated Haematococcus lacustris cells" addresses a very relevant and appropriate topic for this journal. 

The manuscript is well structured, well-written and well-founded. Some minor fixes are suggested below.

RESPONSE: Thank-you for reviewing our submission. We are happy you were positive about content!

Corrections needed:

line 26 - Keywords: Haematococcus lacustris, desiccation tolerance, reactive oxygen species, antioxidants, ...

RESPONSE: We added a semicolon between keywords

line 46 - 25 °C, below a threshold around 44-49% RH, the cytoplasm solidifies into a glassy state,

RESPONSE:  We added 'and' between "25 °C" and "below a threshold around 44-49%"

line 98 - ... and melt. Therefore, cells were equilibrated to 50% RH at 20 °C, which corresponded

RESPONSE:  The sentence was modified to "cells were equilibrated to 50% RH and 20°C, (conditions that lead the cytoplasmic aqueous phase to form a glassy state) before storage in the dark for up to three years at temperatures between −80 °C and 20 °C."

line 257 - above and below these peaks (20 and −80 °C) and within the range of the lipid melting

RESPONSE:  The sentence was modified to: "Based on this thermal profile, temperatures above and below these peaks (20 °C and −80 °C, respectively), and within the range of the lipid melting peaks (4 °C, −20 °C and −50 °C), were used for long term storage of cells in Experiment 1."

line 270 - 3.2. Temperature dependency of ageing reactions in desiccated H. lacustris cells (note: "H. lacustris " in italics)

RESPONSE:  Since the heading is in italics, the species name was intentionally left non italic. 

line 284 - storage at (b) 20 °C, (c) 4 °C, and (d) −20 °C, on a false colour scale shown below the images.

RESPONSE:   The text was changed to "Fv/Fm values from (a), but as images of individual cells using a false-colour scale (below), after 1 year in storage at (b) 20°C, (c) 4°C, and (d) −20°C."

line 288 - for cells stored at 20 and 4 °C, respectively. Ageing at these conditions was significant

RESPONSE:   We changed the text to "Loss of Fv/Fm under these conditions was significant" and deleted the reference to years for clarity

line 308 - both –50 and –80 °C, leading to a slope with smaller correlation coefficient, may be an ...

RESPONSE:   The sentence was changed to "Differences in slope coefficients due to the similar ageing rate at –50 °C and –80 °C may be an artefact of the calculations..."

line 393 - rapidly fell as viability was lost (Fig. S4). In contrast, at 80% RH, γ-glutamyl-cysteine

RESPONSE:  The sentence was changed to "However, with longer time, and similar to GSH concentrations, γ-glutamyl-cysteine concentrations rapidly fell as an increasing number of cells lost viability"